# TRNGs from Pre-Formed ReRAM Arrays

Bertrand Cambou [1,*], Donald Telesca [2], Sareh Assiri [1], Michael Garrett [1], Saloni Jain [1] and Michael Partridge [1]

[1]  College of Engineering Informatics and Applied Sciences, Northern Arizona University, Flagstaff, AZ 86011, USA; sa2363@nau.edu (S.A.); mlg238@nau.edu (M.G.); sj779@nau.edu (S.J.); mcp292@nau.edu (M.P.)

[2]  Air Force Research Laboratory, Information Directorate, Rome, NY 13441, USA; Donald.telesca@us.af.mil

*  Correspondence: bertrand.cambou@nau.edu; Tel.: +1-928-523-7824

**Abstract:** Schemes generating cryptographic keys from arrays of pre-formed Resistive Random Access (ReRAM) cells, called memristors, can also be used for the design of fast true random number generators (TRNG's) of exceptional quality, while consuming low levels of electric power. Natural randomness is formed in the large stochastic cell-to-cell variations in resistance values at low injected currents in the pre-formed range. The proposed TRNG scheme can be designed with three interconnected blocks: (i) a pseudo-random number generator that acts as an extended output function to generate a stream of addresses pointing randomly at the array of ReRAM cells; (ii) a method to read the resistance values of these cells with a low injected current, and to convert the values into a stream of random bits; and, if needed, (iii) a method to further enhance the randomness of this stream such as mathematical, Boolean, and cryptographic algorithms. The natural stochastic properties of the ReRAM cells in the pre-forming range, at low currents, have been analyzed and demonstrated by measuring a statistically significant number of cells. Various implementations of the TRNGs with ReRAM arrays are presented in this paper.

**Keywords:** random number generation; resistive memories; cryptographic systems; unclonable functions; low power

## 1. Introduction

Random numbers play an essential function in cyber security infrastructures. They are absolutely critical in encryption, but are also required in a variety of security scenarios:

Key generation for various algorithms (symmetric, asymmetric, MACs) and protocols (SSL/TLH, SSH, WiFi, LTE, IPsec, etc.):

-  private keys for digital signature algorithms;
-  values to be used in entity authentication mechanisms;
-  values to be used in key establishment protocols;
-  chip manufacturing (seeding device unique and platform keys);
-  initial values (for encryption and MAC algorithms, TCP packet values, etc.);
-  PIN and password generation;
-  nonce generation and initial counter values for various cryptographic functions;
-  and challenges used for protocol authentication exchanges.

In most, if not all, cryptographic systems, the quality of the random numbers used directly determines the security strength of the system, and influences how difficult it is to attack the system. This is not a typical assessment of the strength of a security system where power consumption and bit generation speed, rather than the randomness of the bits generated, are the standard concerns of system designers. Consider that no matter how strong a cryptographic algorithm is, the encrypted message will be completely exposed to adversaries who have knowledge of the algorithm. However, when the algorithm is combined with random numbers, it becomes extremely difficult to compromise the encrypted message, as in the simple one-time pad proposed by Frank Miller. In his book

on telegraphic code, published in 1882, Miller proposed encrypting messages by shifting each letter in the message by a random number of places, resulting in a string of gibberish. Despite the relatively simple algorithm, this one-time pad becomes unbreakable with the addition of truly random numbers.Random number generation is the process which generates a sequence of numbers. True random number generation is expected to yield a sequence that cannot be reasonably predicted, which is better than by random chance. True randomness, however, is exceedingly difficult to achieve. One method to achieve this level of randomness is the measurement of some physical phenomenon from which sufficient entropy can be harvested. This can include the noise produced by a current flowing in a transistor, atmospheric noise, thermal noise or the time between radioactive decay events. Many other sources of entropy such as the ones observed in jitter ring oscillators, metastable states or optical quantum effects are used in working systems. They produce random bit rates up to 10 Gbits per second for optical sources. It is not the intent of the authors to list all the possible sources of entropy to design TRNGs, our focus is to study the applicability of pre-formed ReRAMs for TRNG, in addition to their use to design PUFs. The entropy signal must be conditioned to compensate for a potential bias in the measurement process. A hardware random number generator based on the measurement of physical parameters is also capable of delivering statistically randomness as well as in-deterministic numbers, meeting the essential requirements for secure cryptographic infrastructures. Therefore, a high quality, hardware-based random number generator is considered to be fundamental for delivering the true randomness required for a high level of information security.

The objective of the work presented in this paper is be able to design TRNGs with ReRAM-based PUFs. We wish to be able combine both the TRNGs and the PUFs in cryptographic protocols securing networks for client devices. The difficulty of the approach is in being able to handle conflicting objectives: ReRAM arrays tend to be extremely stable, which is desirable to design reliable key generating PUFs, while the cell-to-cell variations are high and random, representing an opportunity to design TRNGs. The research question is to find a protocol as light as possible to randomly address the cells in the array, and further to enhance randomness after generation from the array of the stream of random numbers. It was then necessary to explore a wide range of options, while quantifying randomness. The analysis in detail of the randomness of the suggested scheme is not a focus of this effort, and will be only conducted on a smaller number of selected methods. We are also departing from prior works that operate ReRAMs at relatively high electric power to form conductive filaments, the schemes here proposed are operating at extremely low power, the "pre-forming" range. The TRNGs are exploiting the natural sources of randomness that are observed in such pre-forming ranges, while making side channel attacks more difficult by operating well below the electrical noise levels. The paper is organized in the following way:

(Section 2): Presents prior works, based on a literature review, of the known methods to design random number generators. At first, we discuss how pseudo random number generators (PRNGs) can be combined with TRNGs to enhance randomness. We secondly present how arrays of memory cells are excellent candidates to design TRNGs due to the natural randomness of each cell that can act independently from the others. Finally, we study how physical unclonable functions (PUFs), that are used for hardware authentication and secret key generation, can also be used to design TRNGs.

(Section 3): In preparation of the design of the TRNGs, this section presents experimental results showing the randomness of ReRAM arrays operating in the pre-forming range. Small electric currents, between 50 nA and 200 nA, are injected in the ReRAM cells to measure the resistance values, which vary randomly cell-to-cell. The possible sources of natural randomness are investigated by characterizing cells with various sizes of active areas.

(Section 4): Presents several methods to design TRNGs with pre-formed ReRAM arrays. In the first version, the resistance values of randomly selected cells are compared

to a reference value close to the median. In the second version, the resistance value of two cells randomly selected are compared. The pseudo codes are presented, as well as the experimental analysis of the levels of randomness.

(Section 5): We propose the addition of post-processing operations, such as the XORing of portions of the random numbers, to further enhance the randomness of the TRNGs. However, the experimental analysis demonstrates that when the natural randomness of pre-formed ReRAM-based TRNGs is good enough, such operations are not needed.

## 2. Background Information on PRNGs and TRNGs

### 2.1. PRNGs versus TRNGs

The PRNGs are widely documented, and available, often they rely on mathematical methods [1–4]. Some PRNGs do not offer enough protection against sophisticated opponents, therefore additional methods to enhance randomness are needed such as the ones based on chaos, and TRNGs [5,6]. The algorithms for PRNGs are often known, so opponents can independently generate long random numbers, and compare them to the ones generated by the PRNGs. Relatively simple matching algorithms can then be used by the adversary to understand the use of the PRNG. In 2018, an article from the National Security Agency (NSA) stated that "the use of PRNGs can result in little to no security; the generation of quality random numbers is difficult" [7]. Conversely, some authors suspect that perhaps backdoors are inserted in commercial PRNGs by governmental institutions such as NSA [8]. TRNGs rely on the natural randomness of physical elements with high levels of stochasticity [9,10]. Considering that the sources of physical randomness are often mixed with sources of more deterministic parameters showing the "DNA" of the physical elements, it is desirable to mix several sources of randomness together, including the ones generated by mathematical algorithms [11]. For example, the source of randomness can be due to both the variations in time of the physical parameters, as well as the noise and uncertainty due to the measurement of these parameters [12]. The design of TRNGs is difficult, problems such as the elimination of oscillations that can be attacked through Fourier transform, need to be addressed. Randomness can be enhanced by combining the streams generated by unrelated physical elements, and by mixing TRNGs with PRNGs. The other types of attacks that are effective include the injection of noise in the TRNG schemes, which can replace the random numbers by data streams known by the opponents. Remedies include the design of TRNGs that can operate at extremely low power, below the noise level, making most noise injection schemes difficult to master and hide. The randomness is quantified by using tests suited for random number generators such as the one described in the NIST publication 800-22 revision 1a [13]. In the case of TRNGs using randomness enhancement schemes, it is recommended to test the data streams a first time after generation from the physical elements, then a second time after final operation. ISO/IEC 20543:2019, within ISO/IEC 19790 and ISO/IEC 15408, is an example of a document specifying a methodology for the evaluation of multi-step non-deterministic or deterministic random bit generators intended to be used for cryptographic applications [14]. Both NIST and ISO documents are implementation-agnostic, they do not offer specific guidance to designing PRNGs or TRNGs, however "passing" such test suites is usually mandatory for commercial use. In this work, several implementations of ReRAM-based TRNGs are analyzed with the methodology recommended by NIST.

### 2.2. Memory-Based TRNGs

TRNGs can be designed based on memory devices such as SRAMs [15–17], DRAMs [18], Flash RAMs [19], Ferro RAMs [20], STT MRAM [21,22], resistive RAMs and Memristors [23,24]. The opportunity is to leverage the potentially large and stochastic cell-to-cell variations, considering that memory arrays contain extremely high numbers of addressable cells that can behave independently. The sources of randomness of memory-based TRNGs are often due to cell-to-cell microscopic variations during the fabrication process, noise in the measurement schemes, and uncertainty in the cell programming process. It has

been proposed that to sense the electron activity in a memory cell such as flash will offer enough randomness when compared with the sensing of the electron activity of other cells, in particular a reference cell [25]. An example of the electron activity in question is the trapping of electrons inside the floating gates of the flash cells. The resulting stochasticity of the flash technology has been characterized to be acceptable to design TRNGs. The method used in this work, which is preferred by many developers, is to randomly test each cell, and compare the reading to a median value of the entire cell population. When the reading is below the median value, a "0" state is generated from the TRNG, a "1" when the reading is above median value. Large and random cell-to-cell variations are expected. To generate a new stream, the TRNG scheme can, for example, use a PRNG to query a new set of randomly selected addresses of cells in the array. The randomness of the PRNG is then combined with the randomness of the memory array. Precautions are usually necessary. This method will work well when the read cycles are not shadowing the randomness of the physical parameters tested. Solid implementations require the elimination of sources that are overly predictable. For example, the periphery of the array is often predictably distinct from the core of the array due to edge effects; this can be mitigated by eliminating from the population generating random numbers, several rows and columns close to the edge. Other issues that have to be corrected are the potential duplications, and deterministic effects, due to the circuitry driving the cells within the memory arrays. For example, in certain designs the circuitry driving the word lines is shared between pairs of word lines. This last issue can be corrected by using only one word line of each pair for the TRNG. The additional work presented in prior work suggests that the stochasticity associated with the programming of arrays of ReRAM or memristors is high enough to design TRNGs [26–29]. However, most programming methods of these devices are dealing with relatively high electric currents that have the potential to damage the cells. Protections are needed. The stochasticity originating from the ReRAM cells can be confused with the random variations of the environment during protection of the cells. Fault injection in the driving circuitry could force the generation of known data streams, instead of the random numbers expected. In contrast, the schemes presented in this paper operate at an exceptionally low current, typically one hundred times lower, in a range that does not form permanent conductive filaments, resulting in much higher stochasticity, as presented in Section 4.

### 2.3. PUFs for the Design of TRNGs

The randomness of the physical parameters that are exploited for the design of TRNGs, such as the cell-to-cell microscopic variations created during fabrication, can be also used to design physical unclonable functions (PUFs) that generate cryptographic keys from Challenge and Response Pairs [30]. Examples of memory-based PUFs were studied in prior work [31–35]. However, with TRNGs, the random numbers generated at each request should be totally different from the previous ones, while the PUFs should always generate the same keys from the same challenge. The reliability of the PUFs are validated by generating the same keys multiple times, and verifying that they are consistently identical. In both cases, TRNGs or PUFs, the data streams generated as either random numbers or keys should pass the test of randomness presented by NIST and ISO [13,14]. The levels of randomness of a memory-based PUF can be validated by generating long data streams, in the multi-megabits range consisting of multiple 256-bit long keys generated from the same PUF. The industrial firm Intrinsic-ID, a provider of SRAM-based PUFs for the Internet of Things, has proposed an implementation of TRNGs that exploit the same PUFs [36]. The source of randomness of the SRAM arrays, which is available by powering up the devices, generates seeds that have enough entropy to be used for the generation of random numbers. The cells of SRAM devices are flip-flops that can switch to a "0" or a "1"state after power off-on cycles. Due to the microscopic variations, certain cells always switch back to the same state, others are highly unstable, which is desirable for the design of TRNGs. Two measurements of the same PUF are always slightly different. In order to be applicable for the design of reliable TRNGs, a sufficient number of bits should

change between measurements in an unpredictable way. We independently conducted experimental work to validate such a method with commercially available SRAMs, and confirmed that between 3% and 5% of the array are highly unstable, and are therefore usable for the TRNG. Methods to enhance the randomness of the streams generated by SRAM-PUFs have been proposed [36–38]. One way to design both a TRNG and a PUF from the same memory array is to use a ternary mapping of the arrays, segmenting the cell population between the predictable cells that generate reliable "0" or "1" states, and the cells that are unstable and marked with the ternary state "X". As described in prior work, the TRNGs are designed by using the unstable cells, while the key generating PUFs are designed with the predictable cells [39,40]. The addresses of the unstable cells, representing 1–3% of the entire population, are stored in a look-up table, and are tested to generate the stream of random numbers. Every time they are tested, the stream is different. Additional steps are needed to enhance the randomness of streams generated by such methods as described in previous publications, in which blocks of random bits are XORed together, i.e., addition mod 2 [41,42]. A statistical model of such an XOR operation is that presented to optimize the size of these blocks to reach a certain level of randomness. One of the constraints of this method is the repetitive use of the same cell population, which could slow down the process for large data streams. In contrast, the schemes presented in this paper can use most cells of pre-formed ReRAM arrays, typically 75% of the total population, for the quick generation of larger streams of random numbers. The stream of random bits directly generated from the physical elements are random enough to pass NIST's suite of tests, while additional XORing of blocks further enhances randomness.

## 3. Randomness of Pre-Formed ReRAMs

### 3.1. Variations in Resistance Value Due to the Injection Small Electric Currents

Pristine ReRAM cells have extremely high resistance values, typically higher than 100 MΩ [43–45]. In the pre-forming range, the injection of very small currents, between 1 nA and 1 μA, forces the resistance values of the cells to drop to the 0.1 MΩ to the 20 MΩ range; after current injection, the resistance values return to their original high resistance values of the pristine state around 100 MΩ. The conduction is thereby ephemeral and reversible. The electrical resistance is defined as the ratio of voltage to the electric current. The pre-formed resistance of each ReRAM cell is unique to that cell and depends on their location in the array, and the physical properties that vary due to the variable defect density observed in the active areas of the cells, along with variations in thickness between metallic electrodes. Millions of pre-formed ReRAM cells were characterized at various levels of currents. Their stochasticity is extremely high, as reported in the example of Figure 1, in which the voltage drops across 200 consecutive cells are measured at the current of 200 nA.

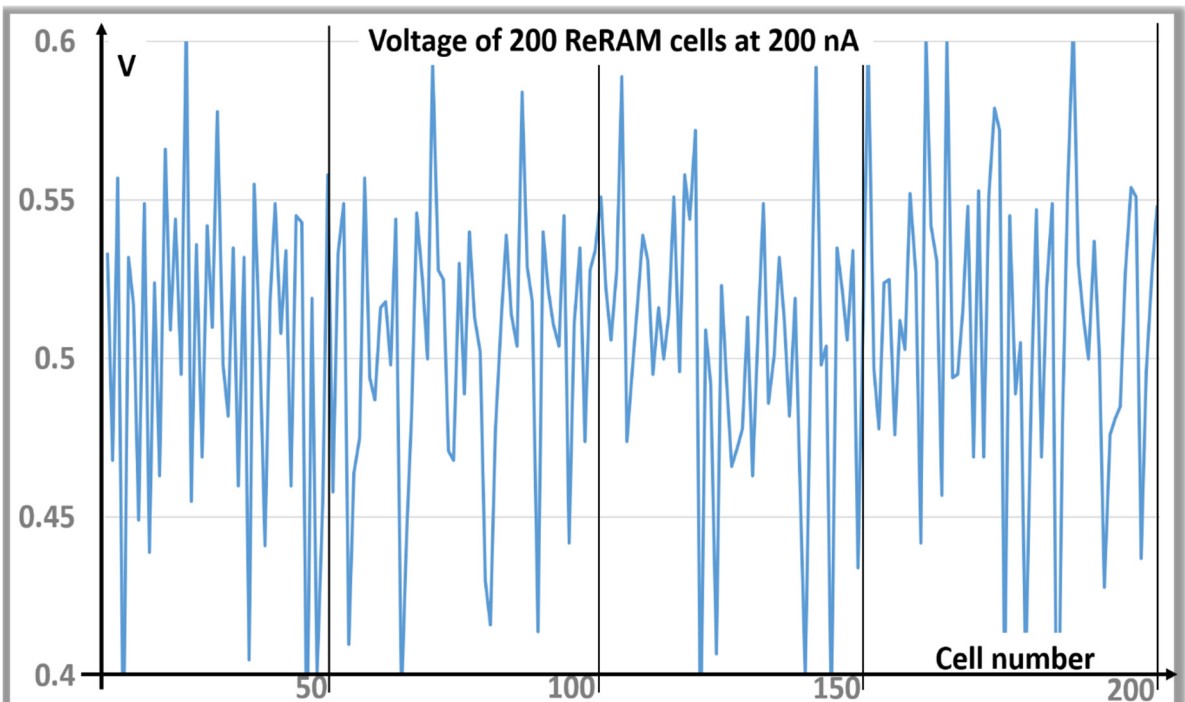

**Figure 1.** Cell-to-cell variations in a pre-formed ReRAM array with constant currents of 200 nA.

For example, the resistance values of a test chip with 2 Mbit ReRAM cells vary randomly in the 2-4 MΩ range. These variations are mainly due to the physical properties of each cell, while the impact of the electrical noise during the read cycles is small. We also noticed that performing repetitive tests does not disturb the cells. Many TRNGs have been designed by exploiting electrical noise, however the levels of randomness of such noise trends tend to be deterministic, as perfectly white noises are hard to find. Based on extensive characterization, it was concluded that the source of stochasticity exploited in this paper, the cell-to-cell variations of pre-formed ReRAMs, does not vary over time, and over several read conditions.

### 3.2. Effect of the Size of the Active Areas on the Levels of Randomness

Figure 2 shows the cell-to-cell variations for various sizes of active areas. These data summarize well the observations made after testing approximately 10,000 cells on the same wafer with the size of the active areas respectively at diameters of 180 nm, 500 nm, 1 μm, and 2 μm. The active areas consist of stacks of two metallic layers separated by the conductive dielectric. The thickness of the dielectrics are approximately constant for all four sizes with values in the 100 nm range. As expected, the resistance values are roughly proportional to the size of the active area, the conduction mechanisms being mainly bulk driven. The levels of randomness that we observed are much lower when the size of the active area is larger.

One of the figures of merit related to the randomness is the standard variation of the parameter, voltage drop or resistance, measured at a constant current injection divided by the median value of the parameter. Such a relative standard deviation is respectively 20% for the cell population at 180 nm, 12% at 500 nm, 6% at 1 μm, and 4% at 2 μm. These variations in randomness can be explained by models based on the density of random defects. Let us assume the following:

i.      when the active area has no defect, the resistance value should be relatively high;

ii.     when one defect is located in the active area, the resistance value should be lower. The resistance reduction could be due to the conductive nature of certain defects;

iii.    when a large number of defects are located in the active area of a particular cell, the addition or subtraction of a few defects does not impact the resistance value anymore.

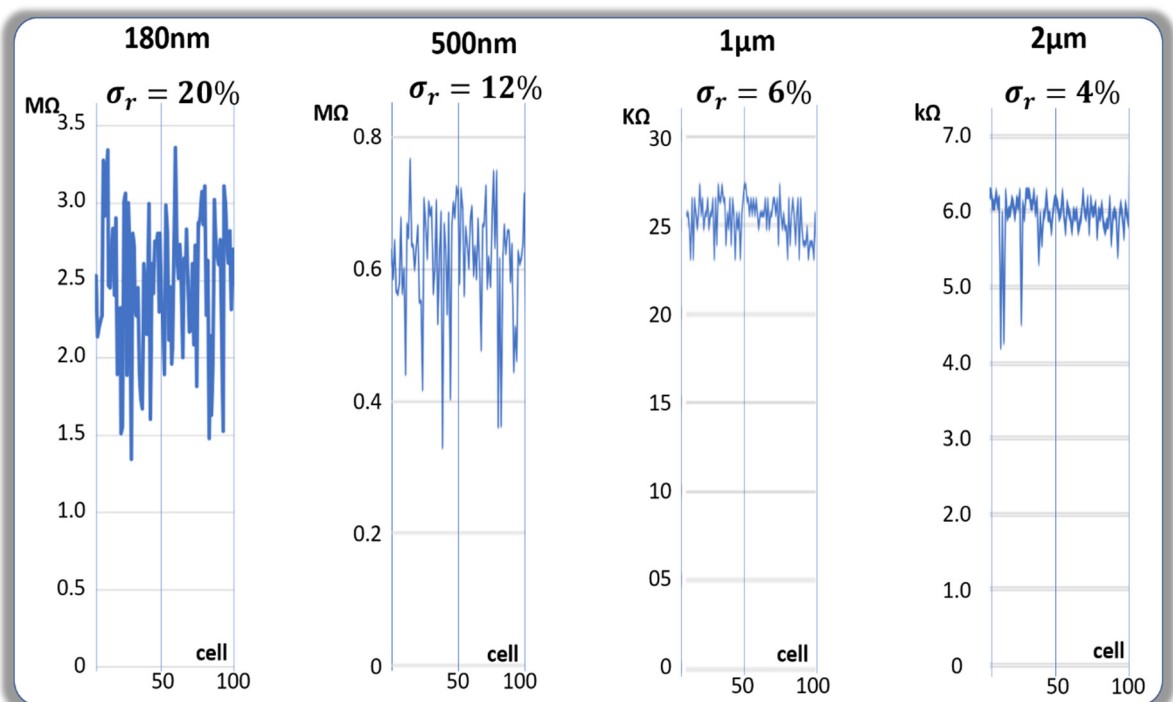

**Figure 2.** Resistance values for 100 cells with 50 nA injected current. The size of the active area of Table 180. nm to 2 μm. The levels of randomness are lower with a larger size.

A set of defects $d_1$, $d_2$, and $d_3$ with defect densities $D_1$, $D_2$, and $D_3$, are affecting the cells having active areas $A_c$ based on the factors $\lambda_1 = A_cD_1$, $\lambda_2 = A_cD_2$, and $\lambda_3 = A_cD_3$. Available statistical models include Poisson, defect cluster, Murphy, Moore, Price, and many others [46–48]. The impact of each type of defect on randomness is low when $\lambda_i << 1$ or $\lambda_i >> 1$. Therefore, it is expected that cells with very small active areas, and cells with very large active areas should exhibit a lower randomness. We anticipate that the manufacturing of very small cells could generate additional defects that can impact randomness. The analytical analysis needed to accurately describe these models is not part of the scope of this study, but will be a part of future work.

## 4. Design of TRNGs with Pre-Formed ReRAM Arrays

### 4.1. Overview of the Architecture of Various TRNG Designs

Exploiting the physical parameters of arrays of ReRAMs described in Section 3 can be considered for the design of key generating PUFs. Such PUFs leverage the resistance value of pre-formed ReRAM cells for the following reasons:

i.  The relative cell-to-cell random variations in resistance values are large, typically in the 50% range, providing high inter-PUF entropy, and unicity;
ii. The variation in resistance values of each cell is small, the intra-PUF variations are low: The relative standard variations of the intra-cell measurements are in the 2% range.

The low intra-cell variations eliminate the possibility to use the same cell in a repetitive way, because of the deterministic behavior of the technology when submitted to the same conditions. The proposed approach to design TRNGs with such PUFs is to randomly select a set of addresses in the ReRAM array, and leverage the cell-to-cell variation in resistance value. As shown in Figure 3, the request for a new random number stream triggers the selection of the set of addresses, which in turn generates the stream of random numbers through pre-formed ReRAM arrays.

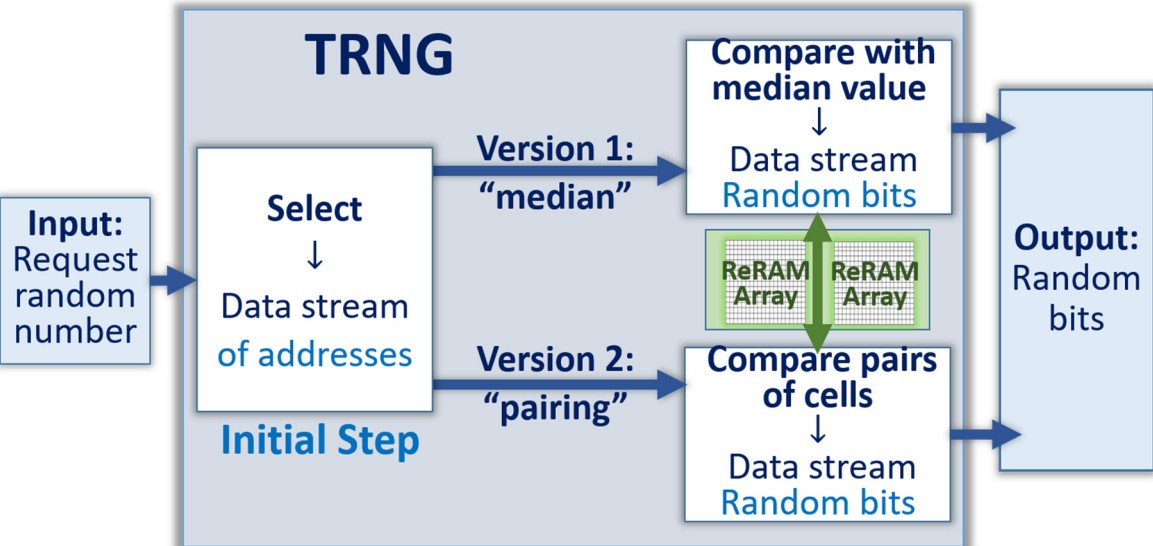

**Figure 3.** Block diagram of the TRNG based on pre-formed ReRAM cells.

In this paper, two versions of TRNGs are studied, the so called "median" version presented in Section 4.2, and the "pairing" version of Section 4.3. The overall architecture of the TRNG is as follows:

(a) Input stage: To request the generation of a new stream of N random bits, for example N = 1000. The only information requested at this step is the length of the stream.

(b) Initial step: To select a stream of at least N addresses in the ReRAM array. For example, if the array has 1 million cells, 20 bits are needed to find the address of one cell in the array. If N = 1000, at least 20 Kbits are needed to find 1000 addresses. This selection can be done for example, by generating a stream of random numbers from a PRNG or a hash function with extended output function (XOF).

(c) Output: To generate N random bits from the N cells selected in the ReRAM arrays with one of the two following methods:

1. Version 1: The median scheme as described in detail in Section 4.2. The stream of N random numbers is generated from the N ReRAM cells selected by the protocol by comparing the resistance values of each cell to a value close to the median. A "0" state is generated if the values are significantly lower than the median value; a "1" state if the values are significantly higher. Most cells of the array are usable for this protocol, with the exception of the cells that are defective, and the ones with values that have resistance values precisely at the median value. These can be excluded from the TRNG.

2. Version 2: The pairing scheme as described in detail in Section 4.3. The stream of N random numbers is generated from the N ReRAM pairs of cells selected by the protocol by comparing the resistance value of pairs of cells. A "0" state is generated if the value of the first cell of each pair is significantly lower than the value of the second cell of the same pair; a "1" state if the value of the first cell is significantly higher. Most pairs of the ReRAM array are usable for this protocol with the exception of the pairs that are defective, and the ones with values that have the two cells with exactly the same resistance value. These can also be excluded from the TRNG.

As presented in the experimental of this section, the measurement of a statistically significant number of pre-formed ReRAM cells, feeding a subset of the suite of tests recommended by NIST, shows that the natural stochasticity of the underlying physical behavior is high enough to pass the tests without additional steps enhancing randomness.

### 4.2. Version 1: Design of TRNGs with the Median Scheme

To generate N random bits, $l$ addresses of cells in the ReRAM array are needed, with $l > N$. Assuming that $f$ bits are needed for each address, a stream of $f \times l$ bits is needed for the median scheme. For example, if the size of the array is 1,048,576 = $2^{20}$, then $f = 20$. A stream longer than N is needed for the protocol to handle the defective cells. The median value $M$ of the resistance values of the entire cell population is also known, as well as some addresses in one array that have values close to the median. The example of Figure 4 shows the implementation with two separate ReRAM arrays, one to find each cell, a second one to find a cell with median value, and an analog circuit comparing the resistance value of the two cells at a given current.

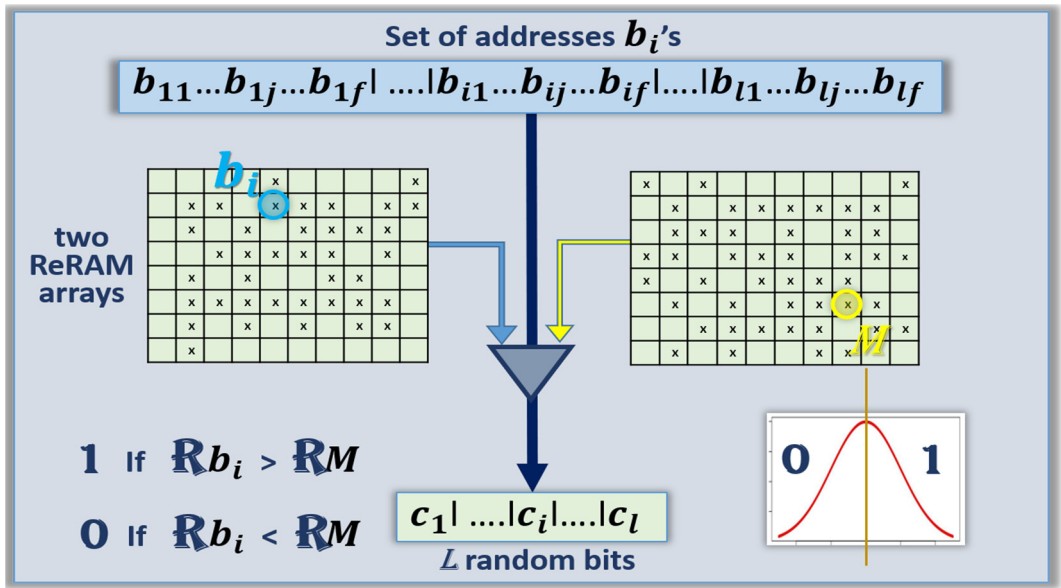

**Figure 4.** Block diagram of a median scheme for pre-formed ReRAM-based TRNGs.

In Figure 4, the cell with median resistance value $M$ is picked on the right side, all other cells are picked on the left side. The simplified protocol is the following:

(a) Input stage: Request N random numbers

(b) Initial step:
PRNG → XOF → random number $l \times f$          ⟵
For $l$ addresses of cells, each of them $f$-bit long, with $l > N$:

$$b_{11} \ldots b_{1j} \ldots b_{1f} \mid \ldots \mid b_{i1} \ldots b_{ij} \ldots b_{if} \mid \ldots \mid b_{l1} \ldots b_{lj} \ldots b_{lf}$$
    Address *1*              Address *i*           Address *l*

(c) Output: Compare the resistance value of each cell to the median value M of the distribution:

    a.      Eliminate the cells that are known to be defective.

    b.      If a cell has a resistance value higher than M → generate a "1".

    c.      If a cell has a resistance value lower than M → generate a "0".

    d.      If a cell has a resistance value precisely at M → ignore.

    e.      If less than N addresses are left → increase the natural number $l$ and iterate ↵

    f.      If more than N addresses are left → keep the first N addresses of the stream.

    g.      Transfer the final data stream of N random number:
$$c_1 1 \ldots .lc_i 1 \ldots .lc_N$$

A pseudo-code used for the implementation of the median scheme generating random numbers from pre-formed ReRAM PUF is shown below Algorithm 1. We studied variations of this implementation that excluded portions of the cell population, such as the cells close to the edge of the array, as well as the cells having resistance values too far from the median M. The impact on randomness is too small.

---

**Algorithm 1:** Generate random number from the **Fuzzy** area close to median in PUF

---

1: **Procedure:** initialize TRNG_SIZE
2:    Call PRNG
3: Arr [] a ← PRNG
4:    FOR every 3byte in a []
   5: convert 3byte to one integer number
6: ENDFOR
7: PUF←READ ReRAM PUF
8: while counter is less than size of TRNG_SIZE
   9: Arr [] RESISTANCE ← GET value from PUF [integer number]
10: ENDLOOP
11: From all RESISTANCE Compute Median
12: Arr [] binary
13: Loop I = 0 to TRNG_SIZE
   14: IF (Median > RESISTANCE [i]) && (RESISTANCE [i] in **Fuzzy area**) THEN
   15: binary[i] ← append (0)
   16: ELSE IF (Median < RESISTANCE [i]) && (RESISTANCE [i] in **Fuzzy area**) THEN
   17: binary[i] ← append (1)
   18: ELSE
   19: continuo
   20: ENDIF
21: ENDLOOP
22: Convert binary to hexadecimal
23: Display TRNG

---

The time to compare the resistance between two ReRAM cells is relatively fast, even with a current injected in the 50 nA range. In the protocol we inject currents in no more than two cells at a given time to avoid disturbing the measurements due to possible sneaky paths. The total current involved is therefore in the nA range. Experimentally we verified that such measurements can be done at a data rate of 10 Kbit per second; 512-bit long random numbers are generated in about 50 ms, which is an acceptable performance for TRNGs. The full characterization and optimization of the latencies of the TRNG is outside the scope of this work.

*4.3. Version 2: Design of TRNGs with the Pairing Scheme*

The generation of N random bits by the pairing scheme requires $2 \times l$ addresses of cells in the array, with $l > N$, which is needed to handle the defective pairs. Assuming that the size of the array is $2^f$, $f$ bits are needed per address, and streams of $2 \times f \times l$ bits are needed for the scheme. The diagram of Figure 5 shows an implementation with two separate ReRAM arrays. The first set of addresses (the $b_i$) is pointing at a first array located on the left side of Figure 5; the second set of addresses (the $\beta_i$) is pointing at the second array located on the right side.

The protocol is the following:

(a)   Input stage: Request N random numbers
(b)   Initial step:
     PRNG → XOF → random number $2 \times l \times f$                                    ⟵
     For $2 \times l$ addresses of cells, each of them $f$-bit long, with $l > N$:

     First cell:        $b_{11} \ldots b_{1j} \ldots b_{1f} | \ldots . | b_{i1} \ldots b_{ij} \ldots b_{if} | \ldots . | b_{l1} \ldots b_{lj} \ldots b_{lf}$
                        Address 1                   Address $i$                   Address $l$
     Second cell:       $\beta_{11} \ldots \beta_{1j} \ldots \beta_{1f} | \ldots . | \beta_{i1} \ldots \beta_{ij} \ldots \beta_{if} | \ldots . | \beta_{l1} \ldots \beta_{lj} \ldots \beta_{lf}$
                        Address 1′                  Address $i$′                  Address $l$′

(c)   Output: Compare the resistance value of each the first cells $b_i$ at address $i$ with their respective matching cells $\beta_i$: at address $i$′:

     a.     Eliminate the cells that are known to be defective.
     b.     If a cell has a resistance value higher than the matching cell → generate a "1".

c.  If a cell has a resistance value lower than the matching cell → generate a "0".
d.  If the two cells of a pair have precisely the same resistance value → ignore.
e.  If less than N random bits are left → increase the natural number *l* and iterate ↵
f.  If more than N random numbers are generated → keep only the N random numbers.
g.  Transfer the final data stream of N random number:

$$c_1 1 \dots .lc_i 1 \dots .lc_N$$

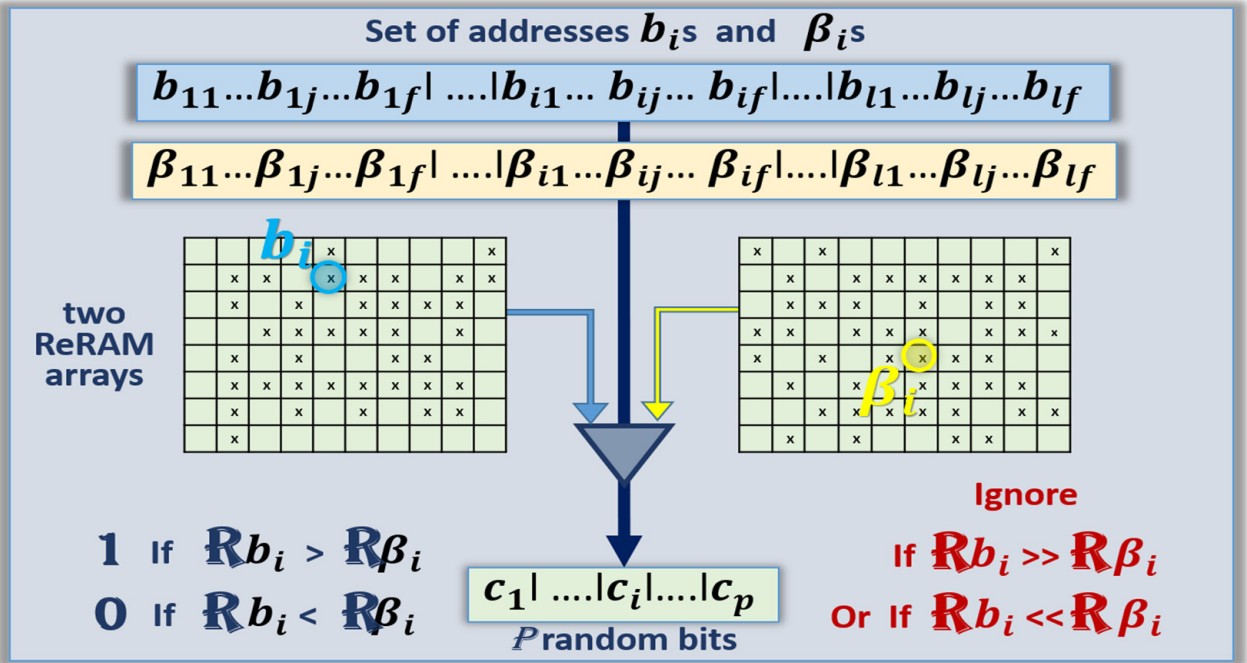

**Figure 5.** Block diagram of a pairing scheme for pre-formed ReRAM-based TRNGs.

A pseudo-code used for the implementation of the pairing scheme generating random number from pre-formed ReRAM PUF is similar than the one presented above for the median scheme with the following differences:

Initial step The PRNG to generate a stream twice as long
Output steps are replaced by the method to compare the pairs of values

### 4.4. Experimental Quantification of the Levels of Randomness

To quantify the levels of randomness of the data streams generated by TRNGs based on arrays of pre-formed ReRAMs, a set of cells was measured at various levels of injected currents from 200 nA to 800 nA. We always inject currents in no more than two cells at a given time to avoid possible sneaky paths. A database of 10 million values was saved, and subjected to both the median and pairing schemes. PRNGs were used to generate the streams of addresses in the database needed in the scheme, and to generate streams of random bits. The randomness of the resulting streams was analyzed by performing 12 tests out of 15 possible tests proposed by NIST. The recommendation of NIST is to use the full 15 tests to develop a fully tested TRNGs, which is mandatory to offer commercially available products. The tool available from NIST allows the selection of a subset of the suite to perform research work, and the comparative analysis of several possible TRNG designs. Some of the tests are long, and require extremely long data streams. The idea is to use a simplified suite of tests to select the most promising configurations of TRNGs during the research phase, then to complete the full suite on the best versions. In this work we tested about 20 different TRNGs, and are reporting the ones teaching us about the way to design TRNGs with pre-formed arrays of ReRAMs. The use of 12 tests is relatively fast

compared with the 15 possible tests; however, the signals were strong enough to detect weaker configurations, and propose remedies.

In analyzing TRNGs, there is often the need to repeat k-times the production of n-bit for the same stream of addresses, therefore generating k x n bits. There is then an expectation that k-bit long sequences are unbiased, zeros and ones should appear with a probability of 0.5. This work is focused on the study of n-bit sequences without repetitions for the following reasons:

i.    We are using PUFs as a source of noise; however, the PUFs need to generate the same response at each address most of the time. This property is mandatory for the design of reliable PUFs with low error rates. For each cell, after k-long sequences, the bias toward zeros or ones should be high, in spite of drifts due to environmental effects and aging.

ii.    However, a very large, and random, cell-to-cell variation is required to design TRNGs. This is the case in ReRAMs operating in the pre-forming range. The cells located at the edge of the arrays could be different to the one located in the center. In order to minimize this effect, we blanked eight rows on each side of the array. The sequences of n-bit considered for this experimental section only used the cells far away from the edge of the arrays.

The protocol developed for the design of TRNGs use PRNGs of variable lengths, combined with hash functions, to randomly address some cells in the array. Typically, we use 10–20 successive bits to find one address. A single bit mismatch of the stream generated with PRNGs will result in a totally different message digest, and a different addressing of the PUFs. With such a protocol, the likelihood that the same sequence of cells is requested twice is extremely low, unless an attack replaces the PRNG by repetitive streams. Presenting methods to prevent such an attack, such as changing the length of the stream of the PRNG, is outside the scope of this paper.

**Median scheme:** Shown in Figure 6 is the summary of the 12 tests performed with the suite of NIST tests on the random numbers generated by the median scheme. Six out of twelve tests are passing with scores higher than 96%; however, six tests are failing. The main reason for such lack of randomness is the unbalance between "0"s and "1"s; we observed 2% more "0"s than "1"s, which is not acceptable. The cell population kept for the TRNG is slightly different to the one used to compute the median value. One way to correct the imbalance is to correct the computation of the median, or to add operations post random number generation from the ReRAM array, as presented in Section 5. The simple XORing of blocks of bits is enough to easily pass the NIST suite of test.

```
RESULTS FOR THE UNIFORMITY OF P-VALUES AND THE PROPORTION OF PASSING SEQUENCES
-----------------------------------------------------------------------------
   generator is <data/reram_stream_x0_10m_median.bin>
-----------------------------------------------------------------------------
 C1  C2  C3  C4  C5  C6  C7  C8  C9 C10  P-VALUE  PROPORTION  STATISTICAL TEST
-----------------------------------------------------------------------------
100   0   0   0   0   0   0   0   0   0 0.000000 *    0/100  *  Frequency
 29  16   9  10   5  10   7   4   6   4 0.000000 *   95/100  *  BlockFrequency
100   0   0   0   0   0   0   0   0   0 0.000000 *    0/100  *  CumulativeSums
100   0   0   0   0   0   0   0   0   0 0.000000 *    0/100  *  CumulativeSums
 94   0   0   2   0   1   1   1   0   1 0.000000 *    7/100  *  Runs
 49  17   9   7   4   7   1   1   0   5 0.000000 *   80/100  *  LongestRun
  9  16   8   9  12  15   6   6  11   8 0.289667     97/100     Rank
  7  10   8  10   5  17   9  15   7  12 0.181557     99/100     FFT
 34  13  11  10   9   6   9   3   4   1 0.000000 *   97/100     ApproximateEntropy
 12  12  11  11   8  13   7   7   7  12 0.798139     97/100     Serial
 14   4   6  15  11  11  11   6  11  11 0.249284     99/100     Serial
  7  13  11  13  12  12  10   7   8   7 0.759756     98/100     LinearComplexity
```

**Figure 6.** NIST-based statistical test of the raw data stream directly generated by TRNGs using the median scheme of pre-formed ReRAM cells. It is a failing NIST suite of randomness tests.

**Pairing scheme:** The PRNGs are used to generate pairs of addresses for the pairing scheme, without excluding the pairs with identical resistance values. Shown in Figure 7 is the summary of the suite of NIST tests. Nine out of twelve tests are passing with scores higher than 96%; however, three tests are still failing with scores respectively at 74%, 76/%, and 80%. Such initial levels of randomness of the streams directly originating from the physical elements are encouraging, and more balanced than the streams generated from the median scheme. Here, we do not have to deal with the problem of centering the median value as statistically 50% of the pairs are either "0"s or "1"s. The post processing presented in Section 5 can enhance randomness to easily pass a subset of 12 NIST tests.

```
RESULTS FOR THE UNIFORMITY OF P-VALUES AND THE PROPORTION OF PASSING SEQUENCES
-------------------------------------------------------------------------------
   generator is <data/reram_stream_x0_10m.bin>
-------------------------------------------------------------------------------
 C1  C2  C3  C4  C5  C6  C7  C8  C9 C10  P-VALUE   PROPORTION  STATISTICAL TEST
-------------------------------------------------------------------------------
 66  15   1   4   2   3   2   0   2   5  0.000000 *   74/100  *  Frequency
  9   8  15  11  13  13   9  10   5   7  0.494392     99/100     BlockFrequency
 64  11   8   3   0   4   2   2   2   4  0.000000 *   80/100  *  CumulativeSums
 63   7   8   6   0   3   2   5   2   4  0.000000 *   77/100  *  CumulativeSums
 10  11   8  10  11  12   7   7  10  14  0.883171     97/100     Runs
 16  11   8  11  11   8   9  11   8   7  0.719747     96/100     LongestRun
  6  18  19   5   7   6   9  11   4  15  0.001201    100/100     Rank
 13   6   8   8  10  12  11  15   6  11  0.534146     99/100     FFT
 15  10  14  11  12   9  10   5   7   7  0.437274     98/100     ApproximateEntropy
  9  14  10   8   5  11  14   8  11  10  0.657933     99/100     Serial
  9  10  12   8  16   8  15   7  12   3  0.137282     98/100     Serial
 14  10  12   8   6   9   8   9  13  11  0.779188     98/100     LinearComplexity
```

**Figure 7.** NIST-based statistical test of the data stream directly generated by TRNGs using the cell pairing scheme of pre-formed ReRAM cells, without removing the pairs with identical resistance values.

When both cells of a pair have resistance values that are exactly the same, a default scheme is needed, which could be acceptable if the occurrence is small enough. In our experiment, as a result of the limited accuracy of the measurements, both cells of 32,000 pairs over 5 million have the same resistance value. This is a relatively small ratio; however, this is high enough to negatively impact NIST's suite of tests of randomness. One way to reduce the ratio is to improve the measurement techniques; however, the latencies needed to accurately measure mega-ohm level resistance values of pre-formed ReRAM cells could be prohibitive. We suggest simply eliminating the pairs with identical resistance values. For this purpose, a circuit with two differential elements was designed.

Figure 8 shows the summary of the subset of 12 NIST tests to quantify the data stream, in which the pairs with identical resistances are removed from the distribution. All 12 tests are passing NIST criteria with an average score of 98%, which is an excellent result. All additional attempts to further enhance the levels of randomness with various PRNGs and XOFs and to generate the stream of addresses resulted in similar satisfactory results. This confirms the excellent stochasticity of the physical parameters driving such TRNGs.

```
RESULTS FOR THE UNIFORMITY OF P-VALUES AND THE PROPORTION OF PASSING SEQUENCES
------------------------------------------------------------------------------
   generator is <data/reram_stream_x0_10m_neq.bin>
------------------------------------------------------------------------------
 C1  C2  C3  C4  C5  C6  C7  C8  C9 C10  P-VALUE  PROPORTION  STATISTICAL TEST
------------------------------------------------------------------------------
 14   7   6  14  12   9   7  12   9  10  0.574903   98/100    Frequency
  6  15   9   8  10  15   7  10   8  12  0.455937   99/100    BlockFrequency
 11  14   7  10  13   7   6   8  10  14  0.534146   97/100    CumulativeSums
 15   7   8   9   5  10   8  13  17   8  0.162606   99/100    CumulativeSums
  6   6   9  15  13  14  10   5  11  11  0.275709   98/100    Runs
 11  11  13   9   9  13   8   6   8  12  0.834308  100/100    LongestRun
 12  11   9   7  13  14  11   3   8  12  0.366918   98/100    Rank
 11  10  13  12  11   4  10   8  11  10  0.779188   99/100    FFT
 12   8  12   8  16   8   7   6  12  11  0.474986   98/100    ApproximateEntropy
 12  11  10  11   6  11  13   9   9   8  0.924076   96/100    Serial
 16   9   9  10   7  11  10  11  11   6  0.678686   97/100    Serial
 12   5  14  13   5   5  12  13  11  10  0.224821   97/100    LinearComplexity
```

**Figure 8.** Testing the data stream generated by TRNGs using the cell pairing scheme of pre-formed ReRAM cells, after removing the pairs with identical resistance values.

## 5. Combining TRNGs with Additional Schemes to Enhance Randomness

### 5.1. Description of the Protocols Providing Additional Randomness

The experimental work presented in Section 4 that is based on 2 Mbit ReRAM arrays is encouraging; however, the levels of randomness could be degraded if smaller arrays are used for the TRNG. We are interested by arrays in the Kbit range, which can be easily integrated in the Internet of Things at a low cost. Even if we do not have any analysis showing that randomness is lower with smaller arrays, we developed additional schemes that should be useful to maintain the performance of the TRNGs.

The approach, as summarized in the block diagram shown in Figure 9, is to further filter the data stream of addresses used for the TRNG, and to expand randomness after generation from the array.

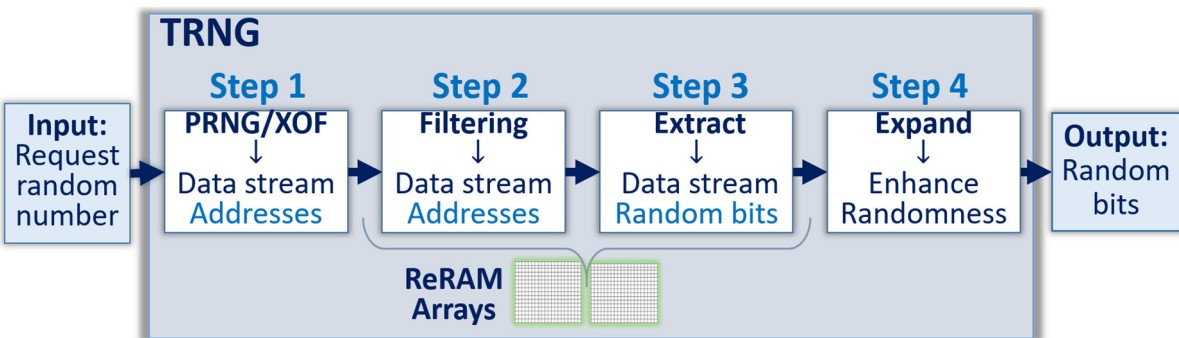

**Figure 9.** Block diagram of an improved TRNG based on pre-formed ReRAM cells and additional steps enhancing randomness.

An example of enhanced protocol is the following:

(1) Step 1: To generate streams of random numbers from the combination of PRNGs, hash functions such as SHA-512, and extended output functions such as shake to select the $l$ addresses needed for the TRNG:

$$b_{11} \ldots b_{1j} \ldots b_{1f} \mid \ldots . \mid b_{i1} \ldots b_{ij} \ldots b_{if} \mid \ldots . \mid b_{l1} \ldots b_{lj} \ldots b_{lf}$$
$$\text{Address } 1 \qquad\qquad \text{Address } I \qquad\qquad \text{Address } l$$

(2) Step 2: To select only the set of addresses in the ReRAM arrays that have the potential to enhance randomness of the TRNG. Example of addresses that are excluded are:

    a.    Defective cells due to catastrophic defects during fabrication.

    b.    Cells with resistance values too far from the median value, see Figure 10. This method excludes the cells that have physical characteristics significantly different from the bulk of the distribution, which could be recognized by the opponents.

    c.    In the case of the median scheme, elimination of the cells that have a resistance value precisely equal to the median value.

    d.    In the case of the pairing scheme, elimination of the pairs with two cells that have the same exact resistance value.

(3) Step 3: Generate N random bits by generating random bits from the ReRAM arrays with one of the methods such as the median scheme, or the pairing scheme as described above in Section 5.

(4) Step 4: Enhance randomness by using additional schemes (see Figure 11), such as various methods used to design PRNGs such as congruent operations in such a way that the flip of only a few bits of the incoming data stream can result in different streams of random numbers.

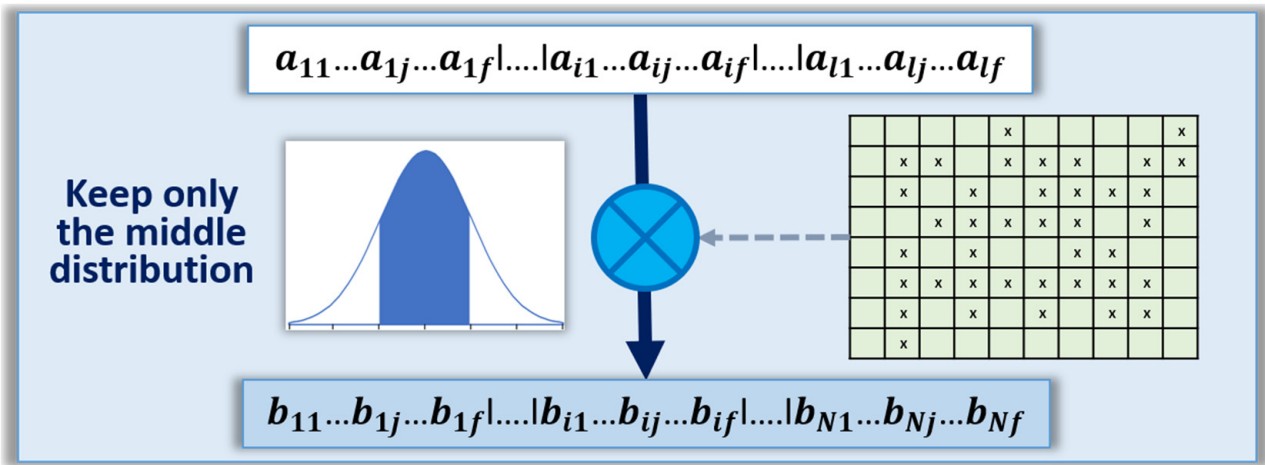

**Figure 10.** Example of the method to enhance TRNGs by filtering the addresses to enhance randomness.

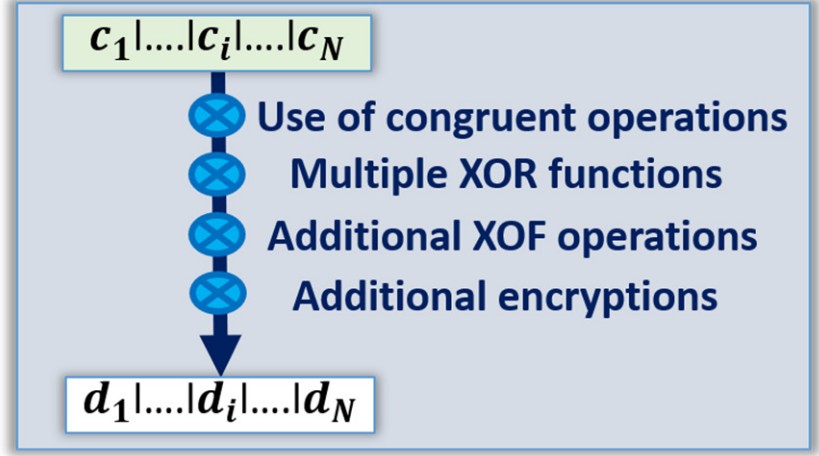

**Figure 11.** Example of methods to expand randomness of the TRNGs.

1.    XORing of the data stream by blocks of bits.

2. Additional hashing and XOF.
3. Additional encryption schemes using the data stream generated from the arrays as both cryptographic keys, and text to encrypt.

Such enhanced TRNGs are expected to deliver higher levels of randomness; however they are slower, use more power, and can be disturbed by malicious fault injections during vulnerable operations. In such cases, less is always better, the addition of steps to enhance randomness has to be minimized. For example, there is little value in further enhancing the randomness of the stream of bits discussed in Section 4.3, it is already passing the subset of 12 NIST tests with scores in the 98% range, which is secure enough for most applications.

*5.2. Experimental Analysis*

In analyzing the effect of additional steps to enhance randomness, the random numbers generated from arrays of pre-formed ReRAM cells in the experimental Section 4.4, were subject to XORing operations.

Median scheme: The random numbers generated with the median scheme presented in Section 4.4 failed six out of twelve NIST tests, largely due to the imbalance between "0"s and "1"s. The additional XORing operation was handled in the following way:

(1) Step 1: To group the streams of random bits by chunks of 7 bits;
(2) Step 2: Add the 7 bits of each chunk modulo 2 to get a zero or a one. This operation is the logical XOR of 7 consecutive bits: if the number of "1"s is odd, the XORing is a "1"; if the number of "1"s is even, the result is a "0". Such an operation enhances randomness [41].

The results, as shown in Figure 12, are excellent. All twelve NIST tests are passing, well within the 96% to 100% range. The six tests failing in Section 4.4, are now passing with scores not differentiable from the six other tests. The XORing operations are fast, and consume low power; however, the latency of such a scheme is by definition seven times slower per bit generated; in order to generate 1000 bits, 7000 bits need to be generated from the ReRAM cells. The optimized scheme can be achieved by XORing chunks of bits that are as small as possible, while having enough levels of randomness to pass the tests.

```
RESULTS FOR THE UNIFORMITY OF P-VALUES AND THE PROPORTION OF PASSING SEQUENCES
-----------------------------------------------------------------------------
   generator is <data/reram_stream_x7_10m_median.bin>
-----------------------------------------------------------------------------
 C1  C2  C3  C4  C5  C6  C7  C8  C9 C10  P-VALUE   PROPORTION  STATISTICAL TEST
-----------------------------------------------------------------------------
 12   7   7   6  14   9   6  12  15  12  0.319084    98/100    Frequency
  7  15   9   5  16  11   7  14   9   7  0.153763   100/100    BlockFrequency
 12   7   8   8  11   8  11   6  13  16  0.455937    98/100    CumulativeSums
 13   8   6  11   4   6  11  15  16  10  0.108791    98/100    CumulativeSums
 16  10  15   5   9   5  13  11   9   7  0.153763    96/100    Runs
 11  10  10  10  12  12  14   8   8   5  0.759756   100/100    LongestRun
 10  12  13   7  11   9  13  13   3   9  0.419021   100/100    Rank
 14   7  13   9  11   3  18   8   7  10  0.062821    99/100    FFT
 21   9  14   9   9  14  10   6   3   5  0.003447    97/100    ApproximateEntropy
 10  17   9  11  10   9   7  12   7   8  0.554420   100/100    Serial
 13  10   9   9  14   7   8  10  11   9  0.897763    98/100    Serial
  9   8   6   8   7  15  14  10  12  11  0.534146    97/100    LinearComplexity
```

**Figure 12.** Test results of the data stream generated with the median scheme, with post processing, the XORing of blocks of 7 bits together. This passes NIST's test of randomness.

Pairing scheme: The first stream of random numbers generated with the pairing scheme, and analyzed in Section 4.4, fails three out of twelve NIST tests due to the presence of pairs having the same resistance value. The sequence presented above with the XORing

by chunks of 7 bits was applied to the resulting random numbers. As is shown in Figure 13, all NIST tests are passing, and the average score is 98.75%. The three tests failing in Section 4.4, are now passing with the score of 100%. While the data stream generated without eliminating the pairs with identical resistance values is failing the frequency test due to the imbalance between "0"s and "1"s, the XORing operation seems to be a simple way to mitigate the problem.

```
RESULTS FOR THE UNIFORMITY OF P-VALUES AND THE PROPORTION OF PASSING SEQUENCES
-----------------------------------------------------------------------------
   generator is <data/reram_stream_x7_10m_raw.bin>
-----------------------------------------------------------------------------
C1  C2  C3  C4  C5  C6  C7  C8  C9 C10  P-VALUE   PROPORTION  STATISTICAL TEST
-----------------------------------------------------------------------------
11  10  14   9   9   9  11  10   9   8  0.978072   100/100    Frequency
 7  10   7  15   9   7  10  12  13  10  0.678686    99/100    BlockFrequency
11  11   8   7  12   9   9  10  12  11  0.978072   100/100    CumulativeSums
10  13   7  11  10   6   6  17   6  14  0.153763   100/100    CumulativeSums
 6   6   9  14  11  12  12  10   8  12  0.678686    98/100    Runs
11   6  12  11  12   9  12  12  10   5  0.739918    98/100    LongestRun
 9  11  13  13   6  10   8  10  11   9  0.897763    99/100    Rank
11   7   8  12  13  11   8   9   9  12  0.924076    99/100    FFT
14  17   7   9   7   9  10  10  10   7  0.401199    98/100    ApproximateEntropy
11   8  12   4   6   5  18  16  11   9  0.026948    98/100    Serial
 7  10   9   8  11   8   9  10  18  10  0.494392    99/100    Serial
12   8   8   7  13  14  12  10   8   8  0.759756    97/100    LinearComplexity
```

**Figure 13.** Statistical test of the random numbers generated by the cell pairing scheme with a XORing by 7 bits.

As shown in Section 4, the streams of random numbers generated with the pairing scheme, and after elimination of the pairs with identical resistance values, are passing all NIST tests with an average score of 98%. In order to evaluate the impact of an additional XORing operations, the resulting streams were subjected to the following operations:

(1) Step 1: To group the streams of random bits by chunks of 11 bits;
(2) Step 2: Add each chunk modulo 2 to get one resulting bit, a zero or a one. This operation is equivalent to XORing 11 bits, if the numbers of "1"s is odd, the resulting bit is a "1", if even the resulting bit is a "0". The levels of randomness increase with the length of the chunks.

The twelve tests, as shown in Figure 14, are now passing with an average score of 99%, which is higher than what was reported without XORing. Ten of the twelve tests score 99%, or higher, which is as good as one can expect considering the length of the data stream generated by the TRNG. However, the 98% score before XORing is already good enough, the score of 99% does not represent a desirable improvement. NIST is recommending a threshold of 96%, not 98%, to avoid levels of randomness that are suspiciously high. Therefore, two implementations of TRNGs are recommended: (i) screening the cells or pairs to balance in "0"s and "1"s, or (ii) adding post processing operations such as XORing chunk of bits. Combining the two methods is not necessarily better, however excellent statistical properties of TRNGs are usually the result of a combination of deterministic processes, non-deterministic process, and chaotic noise. The results presented in this experimental section seem to indicate that perhaps the post-processing operation could be excluded, which could improve both latencies, and more importantly exposure to fault injection in the post-processing operations. Additional research work is absolutely needed to make such a recommendation; the full 15 tests of the suite developed by NIST are then necessary, as well as the generation of much longer streams of random numbers.

```
RESULTS FOR THE UNIFORMITY OF P-VALUES AND THE PROPORTION OF PASSING SEQUENCES
-----------------------------------------------------------------------------
   generator is <data/reram_stream_x11_10m_neq.bin>
-----------------------------------------------------------------------------
 C1  C2  C3  C4  C5  C6  C7  C8  C9 C10   P-VALUE   PROPORTION  STATISTICAL TEST
-----------------------------------------------------------------------------
  9   8  11   5   6  12  13  12  13  11  0.595549    99/100    Frequency
  9   9   7  17  10  10   8   6  13  11  0.437274    99/100    BlockFrequency
 10   8   4   9  19   7   4  14  12  13  0.020548    99/100    CumulativeSums
 11  12   9   6   8  11   7  17   9  10  0.474986    99/100    CumulativeSums
  9  10  13   6   6   8   8  14  13  13  0.494392    99/100    Runs
  9   8  10  13   9   9  12   9  11  10  0.987896   100/100    LongestRun
  5  11  12  12  16   4  13  10   6  11  0.153763   100/100    Rank
  7  10  10   8  14  11  11  11   9   9  0.946308   100/100    FFT
 12   7   7  13   7  14   6  12  13   9  0.474986    99/100    ApproximateEntropy
 16  10   9   9   8  11   5  11   7  14  0.401199    97/100    Serial
 12  18  13   7   6   8   5   8   9  14  0.085587    98/100    Serial
 11  17   7  13   7   8  12   9   6  10  0.334538    99/100    LinearComplexity
```

**Figure 14.** Statistical test of the data stream generated with pairing scheme, elimination of the pairs with identical resistance values, and a XORing by chunk of 11 bits.

## 6. Conclusions and Future Work

The design of various TRNGs based on the use of pre-formed arrays of ReRAM cells has been investigated. The cells were characterized at extremely low injected currents, between 50 nA and 200 nA, well below the levels needed to form partially conductive filaments, in a range that does not create detectable changes in physical properties. The cell-to-cell variations in resistance values are remarkably large and random, which is opening up the possibility to design TRNGs. The statistical analysis shows that the levels of randomness are higher when the active areas of the cells is reduced from a 2 μm diameter to 180 nm. We concluded that the root cause of the cell-to-cell randomness is most likely due to random defects; however, this needs to be confirmed by additional physical analysis that we intend to conduct as future work.

We proposed two possible methods to exploit the stochasticity of the resistance values of pre-formed ReRAM cells to design TRNGs. In the first method, a group of cells is randomly selected, tested, and their resistance values are compared to the median value of the entire population. When the resistance value of a cell is lower than the median, a "0" state is generated; when the resistance value is higher than the median, a "1" state is generated. In the second method, a group of pairs of cells are randomly selected. When the resistance value of the first cell of a pair is lower than the value of the second cell, a "0" state is generated; a "1" state is generated in the opposite configuration. The latencies to perform the comparative measurements of both methods are less than 100 μs per bit; 512-bit long random numbers are generated in about 50 ms at a data rate of 10 Kbit per second. As part of future work, we are currently developing the integrated circuitry to reduce the latencies by two orders of magnitude. The long-term objective is to reduce these latencies to 1 μs per bit, and to increase the data rate of the TRNG to 1 Mbit per second. Various schemes to enhance the randomness of the TRNGs were considered:

o   PRNGs, hashing functions, and XOFs to randomly select the group of cells within the ReRAM arrays generating the random numbers.

o   Eliminating certain cells that can reduce randomness and protect the arrays.

o   Post-processing operations: additional hashing, XORing, and encryption.

The streams of random numbers generated directly from 2 Mbit pre-formed ReRAM arrays are passing a subset of 12 NIST tests. To pass these tests, the distribution of pairs of cells that have their resistance values precisely identical were removed from the distribution. Additional XOR operations further enhance randomness and result in data streams

passing NIST tests; however, such steps may not be necessary considering that the levels of randomness are good enough without them. As future work, we intend to study the most promising configurations of TRNGs with the full suite of 15 NIST tests. We will also study ways to reduce the size of the ReRAM arrays to reduce costs for implementation in light IoTs. We anticipate that post-processing enhancing methods could then be necessary.

The work to implement the proposed TRNG using pre-formed ReRAM arrays is not underestimated by the authors. One of the major tasks will be to anticipate potential attacks, and implement mitigations. For example, pointing a laser at one of the two blocks can increase its temperature, thereby reducing the resistances, and degrade the stability of the TRNG. We are considering the use of 16 blocks a single memory devices. At a given time two blocks out of 16 will be randomly used to generate the random numbers. Statistically each block is only used 1/8 of the time. Pointing at one block will then be difficult, as the heat will quickly diffuse to the adjacent blocks. Other remedies to this type of attack is to use some of the ReRAM cells with known resistance as temperature-sensing elements. To support the research effort, a custom circuit was fabricated that can exploit the ReRAM-based PUF both for random numbers and cryptographic key generation. The size of the array is only a 4 Kbit, it remains to be proven that the TRNG can then pass the NIST test suite.

**Author Contributions:** Conceptualization, B.C.; methodology, M.G., B.C.; software, S.A., M.G., S.J., M.P.; validation, S.A., M.G.; formal analysis, M.G., S.A., S.J., M.P.; investigation, M.G., S.A., S.J., M.P.; resources, D.T., B.C.; data curation, S.A., M.G., S.J., M.P.; writing—original draft preparation, B.C., D.T., S.A.; writing—review and editing, S.J., M.P.; visualization, M.G., S.A., S.J., M.P.; supervision, B.C., D.T.; project administration, B.C., D.T.; funding acquisition, D.T., B.C. All authors have read and agreed to the published version of the manuscript.

**Funding:** This material is based upon the work funded by the Information Directorate under AFRL award number FA8750-19-2-0503.

**Data Availability Statement:** Not applicable.

**Acknowledgments:** The authors are thanking the staff, students and faculty from Northern Arizona University (NAU), in particular Ian Burke who is managing NAU's cybersecurity lab, Taylor Wilson, and Morgan Riggs. We are also thanking the professionals of the Air Force Research laboratory (AFRL) of Rome, New York (US), who supported this effort. (a) Contractor acknowledges Government's support in the publication of this paper. This material is partially based upon the work funded by the Information Directorate, under the Air Force Research Laboratory (AFRL); (b) Any opinions, findings and conclusions or recommendations expressed in this material are those of the author(s) and do not necessarily reflect the views of AFRL.

**Conflicts of Interest:** The authors declare no conflict of interest. The funders had no role in the design of the study; in the collection, analyses, or interpretation of data; in the writing of the manuscript, or in the decision to publish the results.

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
