# Peer review of "TRNGs from Pre-Formed ReRAM Arrays"

_cryptography, doi:10.3390/cryptography5010008_

Round 1

Reviewer 1 Report

The paper concerns a TRNG design with arrays of ReRAM cells, also called memristors. The authors propose to use tamper sensitive ternary ReRAM-based PUFs that will be presented during SAI computing conference in July 2021. The proposed approach to design TRNG with such PUFs is to randomly select a set of addresses in the ReRAM array and leverage the cell-to-cell variation in resistance value. The source of random addresses is a pseudo-random generator. The request for a new random number stream triggers the selection of the set of addresses, which in turn generates the stream of random numbers through pre-formed ReRAM arrays.

In the reviewer’s opinion the subject is important and the paper is innovative but cannot be published in the current form. The reasons are the following:

  • The paper uses an article that will be published in the future. Article [49] is not available to the reviewer.
  • The authors’ contribution is not sufficiently clear. The main points of the authors’ contribution should be presented in the Introduction. General statements like “In this paper, the authors are presenting a research effort to design TRNGs with arrays of ReRAM cells, also called memristors. ….” are insufficient.
  • The results of statistical tests are incomplete. The test suite proposed by NIST in Rev. 1a, originally published in 2010, contains 15 groups of tests. The authors present results of 12 tests.
  • The proposed TRNG uses a true randomness offered by ReRAM cells, pseudo-random streams of addresses produced by a PRNG and additional mathematical transformations, called post-processing, that improve statistical properties of raw sequences. It is known that excellent statistical properties can be the result of a deterministic process, non-deterministic process, deterministic chaos or a combination of mentioned. To exclude significant impact of the deterministic component on the results of statistical properties of output sequences, additional research work is required. Repeating the production of n-bit sequences k-times for the same streams of addresses a matrix of k x n bits is obtained. The authors showed that n-bit output sequences have excellent statistical properties but did not prove that k-bit sequences (the number of k-bit sequences is equal to n) are unbiased, i.e., that zeros and ones in k-bit sequences appear with the same probability. When perfect statistical properties are the result of a deterministic process, k-bit sequence contains only zeros or ones for i=1,2,…,n. When perfect statistical properties are the result of a non-deterministic process or the chaos phenomenon with sufficiently high topological entropy, k-bit sequences are unbiased, i.e., zeros and ones appear with probability 0.5. Significant deviations from this value are dangerous because they open new doors to different attacks and manipulations.
  • Sources of entropy are not limited to “the noise produced by current flowing in a transistor, atmospheric noise, thermal noise or the time between radioactive decay events” but include also jitter in ring oscillators, metastable states or optical quantum effects. All of these sources are used in working systems, producing random bits with bit rates from single Mbit/s to more than 10 Gbit/s for optical sources.

Author Response

Dear Colleague,

Thank you very much for your valuable feedback. Please find enclosed a pdf documents describing what we did step-by-step in response of your review.  This is an extremely important paper for us representing hard work. Thanks to your comments, this is a much improved paper.

Respectfully,

The authors.

Reviewer 2 Report

In this paper pre-formed ReRAM array of cells are evaluated as TRNGs. Two methodologies are applied based on the comparison of drop voltages induced by the injection of a small current in the pre-formed ReRAMs. First, the comparison against a median value is used and second the comparison between pairs of cells. NIST tests are assessed and it is demonstrated the good quality of bit streams obtained from this methodology.
Overall the presentation of the work is excellent, considering its structure, writing and illustration of figures and tables. Also, it is worth mentioning the profusion of bibliography used.
Considering the final version I would like to suggest the following minor things:
1) Authors indicate that a small current (50 to 200nA) is injected. I assume that this is for one cell. However, in 1Mb memory this current should be multiplied by the number of cells in a row or column? What would be the total current involved in the drop voltage measurement?
2) Might the stability of this TRNG be broken by heating one of the two memory blocs, e.g. with a laser? How could this be prevented?
3) Ln 117: I would eliminate the comma after problems
4) Ln 210: I guess "calls" should be "cells"
5) Ln 263: "currant"
6) Ln 435: a set "of" cells

Author Response

Dear Colleague,

Thank you very much for your encouraging, and constructive feedback.  In particular, your suggested attack with a laser is extremely interesting. We incorporated a section at the end of our paper suggesting counter measures.

Respectfully,

The authors

Round 2

Reviewer 1 Report

The authors have completed the paper revision according to the reviewers' comments. It is suggested to be accepted for publication.